# PeerJ

# Chronologically sampled flight feathers permits recognition of individual molt-migrants due to varying protein sources

Sievert Rohwer[1], Anthony D. Fox[2], Thomas Daniel[3] and Jeffrey F. Kelly[4]

[1] Department of Biology and Burke Museum, University of Washington, Seattle, WA, USA
[2] Department of Bioscience, University of Aarhus, Kalø, Grenåvej, Rønde, Denmark
[3] Department of Biology, University of Washington, Seattle, WA, USA
[4] Oklahoma Biological Survey and Department of Biology, University of Oklahoma, Norman, OK, USA

## ABSTRACT

This is a proof of concept paper based on chronological samples of growing feathers from geese thought to be molt-migrants. When molt-migrant birds initiate molt shortly after migrating to a new isoscape, isotope values measured along the length of their feathers should change continuously. To assess long-term changes and daily cycling in $\delta^{15}$N and $\delta^{13}$C values, we serially sampled a growing primary from three presumed molt-migrant geese. Two showed changing $\delta^{15}$N signatures along the length of their growing primary, indicating they were molt-migrants, while the third, presumably a resident, showed no change. We then resampled these feathers at closer intervals for evidence of the predicted diel cycle in the use of exogenous and endogenous protein for feather growth, generated by the diel feeding cycle of these geese. As predicted, a periodicity of ca. 24 h in $\delta^{15}$N values was found along the primary of the two equilibrating geese, but not in the other goose that was probably a resident. Our results demonstrate that chronological sampling along the length of individual primaries holds great potential for identifying individuals that are molt-migrants.

Corresponding author
Sievert Rohwer, rohwer@uw.edu

## INTRODUCTION

Molt-migrants require time for isotopes in their endogenous protein reserves to come into equilibrium with local isoscapes (*Martinez del Rio & Anderson-Sprecher, 2008*). Thus, feathers grown during this period of equilibration should show a steady change in isotopic signature along their length. This follows because feathers grow at a more or less constant rate throughout the 24-hour cycle (*Murphy & King, 1986*; *Schieltz & Murphy, 1995*; *Lillie& Wang, 1940*); thus, when exogenous protein from local foraging is exhausted, endogenous protein reserves supply the protein needed to build feathers during daily periods of fasting (*Murphy & King, 1990*).

These facts generate two predictions that help identify molt-migrants that initiate feather growth shortly after moving to a new isoscape. First, molt-migrants that begin

molt before equilibrating to a new isoscape should show a steady change in isotope signature along the length of their flight feathers. This prediction is easily tested with course sampling along the length of flight feathers. Second, if local exogenous protein sources are exhausted between bouts of feeding, fine sampling along feathers, should reveal a roughly 24-hour periodicity representing the alternating use of exogenous and endogenous protein for feather construction. The amplitude of this cycle is likely driven by the size of the protein pool in the blood relative to the amount of protein being withdrawn for feather synthesis. Depending on the fraction of blood protein used for feather synthesis, daily feeding and fasting should generate measurable (if sometimes small) changes.

Greylag Geese (*Anser anser*) that breed in terrestrial, freshwater habitats in Sweden are known to migrate in late summer to the maritime saltmarshes of the island of Saltholm (55°38′N 12°45′E, *Nilsson, Kahlert & Persson, 2001*) in Denmark where they undergo their annual molt while grazing on saltmarsh plants (*Fox et al., 1995*; *Fox, Kahlert & Ettrup, 1998*). Compared to freshwater environments, marine environments are known to have very different $\delta^{15}$N signatures, and they may have slightly different $\delta^{13}$C signatures. Correspondingly, stable isotope values in growing flight feathers of Greylag Geese were intermediate between birds on a terrestrial plant diet and those on a saltmarsh diet (*Fox, Hobson & Kahlert, 2009*).

To explore expected changes in $\delta^{15}$N and $\delta^{13}$C along the length of primaries and to examine the potential for 24-hour cycling in the use of exogenous and endogenous protein for feather growth, we serially sampled a single primary feathers from each of three Greylag Geese from Saltholm; these feathers (and no others) were available from the study of *Fox, Hobson & Kahlert (2009)*. Because feathers are generated from tip to base and composed of non-living keratin, primaries grown by geese that migrated to Saltholm for their molt should change chronologically from more positive (freshwater) $\delta^{15}$N signatures at the feather tip to more negative (marine) $\delta^{15}$N signatures toward the feather base. Further, because flightless Greylag Geese forage only at night on Saltholm (*Kahlert, Fox & Ettrup, 1996*), a 24-h periodicity in the $\delta^{15}$N signature should be expected in serial samples from the flight feathers of geese that were not in equilibrium with the Saltholm isoscape.

Our results show that serial samples taken along the length of primary flight feathers readily show equilibration toward a new isoscape and, further, that feathers sampled at fine intervals can reveal diel shifts in exogenous and endogenous protein sources used for feather generation. These results clearly demonstrate that the chronological samples available from individual feathers can be used to identify individuals that are molt-migrants. We are aware of just two prior studies of hair or feathers that used chronological samples to examine changes in isotope signatures through time. *Cerling et al. (2006)* used elephant hair to demonstrate the movement of individual African elephants (*Loxodonta africana*) to different foraging locations, and *Church et al. (2006)* used samples along the length of a growing rectrix from a California Condor (*Gymnogyps californianus*) to show a sudden deposition of lead that resulted in its death. Because our results are unavoidably based on feathers from just three geese, their importance lies in demonstrating the value of chronological samples to study molt-migrantion in birds.

## MATERIALS AND METHODS

Feathers were obtained from Greylags in active molt on Saltholm, between the Copenhagen and Swedish Skania coast (*Fox, Hobson & Kahlert, 2009*). The Danish Forest and Nature Agency gave permission to catch and sample the geese and the landowners of Saltholm gave permission to work on the island. The geese were caught under the Copenhagen University Natural History Museum Ringing Permit A600 "DMU-Kalø Ringmærkning."

The geese included in this study were not individually tracked, so we do not know when they arrived at Saltholm, or how soon after arrival they started to molt, or even if they were all molt-migrants to Saltholm. These limitations mean that useful information on periodicity could only come only from birds that showed declining $\delta^{15}$N signatures along the length of the sampled primary, indicating they were molt-migrants to Saltholm. Greylags are large geese that would require considerably more than a month to come into equilibrium with the new isoscape of Saltholm (*Martinez del Rio & Anderson-Sprecher, 2008*). Molting geese in equilibrium with the salt marsh isoscape of Saltholm should show no 24-hour cycling in isotopic signatures; these could be either local, Saltholm, geese (e.g., failed breeders) or salt marsh breeders from a similar isoscape. Although there is variation among individual feather growth rates, Greylag Geese should grow their feathers at approximately 7 mm d$^{-1}$, as inferred from a mean mass of 3,509 g (*Dunning, 2007*) and the allometric relationship between primary growth rate and body size (*Rohwer et al., 2009*).

The feathers used for this study were sampled twice: first at 5 mm intervals from the tip of the primary to measure change in $\delta^{15}$N and $\delta^{13}$C over long time periods, and second, at 1 or 2 mm intervals to assess 24 h periodicity in the protein source used for feather generation. Which primary feather was used should not affect the results of this study, as Greylag Geese, like most waterfowl, lose and replace their flight feathers simultaneously (*Hohman, Ankney & Gordon, 1992*).

Methods for sample preparation and analysis generally follow those in *Paritte & Kelly (2009)*. Feathers were cleaned in dilute detergent followed by repeated rinsing. After air drying the feathers were cleaned again in 2:1 chloroform:methanol and allowed to air dry before processing. Once the feathers were dried, a research technician marked the rachis of each feather from its tip to the base of the growing vane at 1, 2 or 5 mm intervals. The majority of the posterior vane was then cut away, leaving only the few mm closest to the rachis. Then, using the pen marks as a guide, an approximately 200 µg sample of feather vane was cut immediately adjacent to the pen mark. These samples were loaded into tin capsules (3.5 × 5.5 mm) and stored in an elisa plate until they were analyzed.

We analyzed samples in batch sequences of 49 samples and references, referred to as autoruns. Each autorun typically analyzed 39 unknown samples and 8 laboratory reference samples in positions 1, 2, 7, 13, 19, 37, 43, 49. The laboratory reference material was powdered Brown-headed Cowbird feather (*Molothorus ater*), as described in *Kelly et al. (2009)*. Among sample variation in the laboratory reference material was <0.2‰ for both $\delta^{13}$C and $\delta^{15}$N. In addition to this laboratory reference, we ran one sample each of two National Institute of Standards and Technologies NIST reference materials (USGS 40

in autorun position 25 and USGS 41 in autorun position 31). All stable isotope ratios are expressed in standard $\delta$ notation, where $\delta^{13}C$ and $\delta^{15}N$ = [(isotope ratio sample/isotope ratio standard) − 1] * 1,000. Consequently, $\delta^{13}C$ and $\delta^{15}N$ are expressed in parts per thousand (‰) deviations from a standard, which was Vienna Pee Dee Belemnite for $\delta^{13}C$ and air for $\delta^{15}N$. Isotope ratios were measured at the University of Oklahoma using a Thermo Finnigan Delta V isotope ratio mass spectrometer connected to a CosTech elemental analyzer.

For each autorun we corrected all measurements for instrumental drift between the first and last sample. Instrumental drift corrections were based on the slopes of best-fit lines for $\delta^{13}C$ and $\delta^{15}N$ values regressed against analysis time of references within each autorun. A slope was calculated for the cowbird standard in the run and this slope was used as the drift correction coefficient.

To determine if there was evidence of 24-hour cycling in the $\delta^{15}N$ values along the length of the primaries, we used custom Matlab code to perform an autocorrelation analysis after de-trending the data using linear regression and removing the mean. We used this method to find correlation maxima and minima that reveal periodicity in the $\delta^{15}N$ values. Using additional custom Matlab code, we then developed a bootstrap method to test for the statistical significance of having autocorrelation minima and maxima that correspond to a periodic pattern in isotope values. To do so, we randomly permuted the data for each feather and performed autocorrelation analyses of those permuted values. Out of 10,000 permutations, we asked what fraction of the data had both a minimum less than or equal that observed in our original autocorrelation and a maximum spaced at the appropriate interval.

## RESULTS

### Evidence for equilibration following molt-migration

$\delta^{15}N$ signatures declined over time in two geese (501 and 508, $P < 0.0001$), and remained constant in the third (509, $P = 0.34$, Fig. 1). Both geese that changed did so in a way consistent with the large shift of about 8‰ in the $\delta^{15}N$ isoscapes suggested by the results of *Fox, Hobson & Kahlert (2009)*. The $\delta^{15}N$ was constant, with a mean of 7.6‰, along the primary for goose 509, which is puzzling because this mean is intermediate between the beginning and ending values for the other two geese (Fig. 1). If this individual had been on Saltholm long enough to be in equilibrium with the salt marsh isoscape, then its mean $\delta^{15}N$ value should have been at or below the latest values from the two geese with declining $\delta^{15}N$ values. That its mean $\delta^{15}N$ was considerably higher than the lowest $\delta^{15}N$ values found for the two geese coming into equilibrium (Fig. 1), suggests it was a resident goose that did not feed in the Saltholm salt marshes. Hayfields used by resident Greylag Geese are less subject to marine influence than the saltmarshes where the majority of migrants feed.

The $\delta^{13}C$ signatures of molt-migrant geese should decline if they moved from Sweden to Saltholm for the molt (*Fox, Hobson & Kahlert, 2009*). Yet, the $\delta^{13}C$ signature of goose 501 increased along the length of its primary ($p = 0.0003$), while $\delta^{13}C$ remained constant for 509 ($p = 0.09$) and 508 ($p = 0.46$, Fig. 1). The decline in $\delta^{15}N$

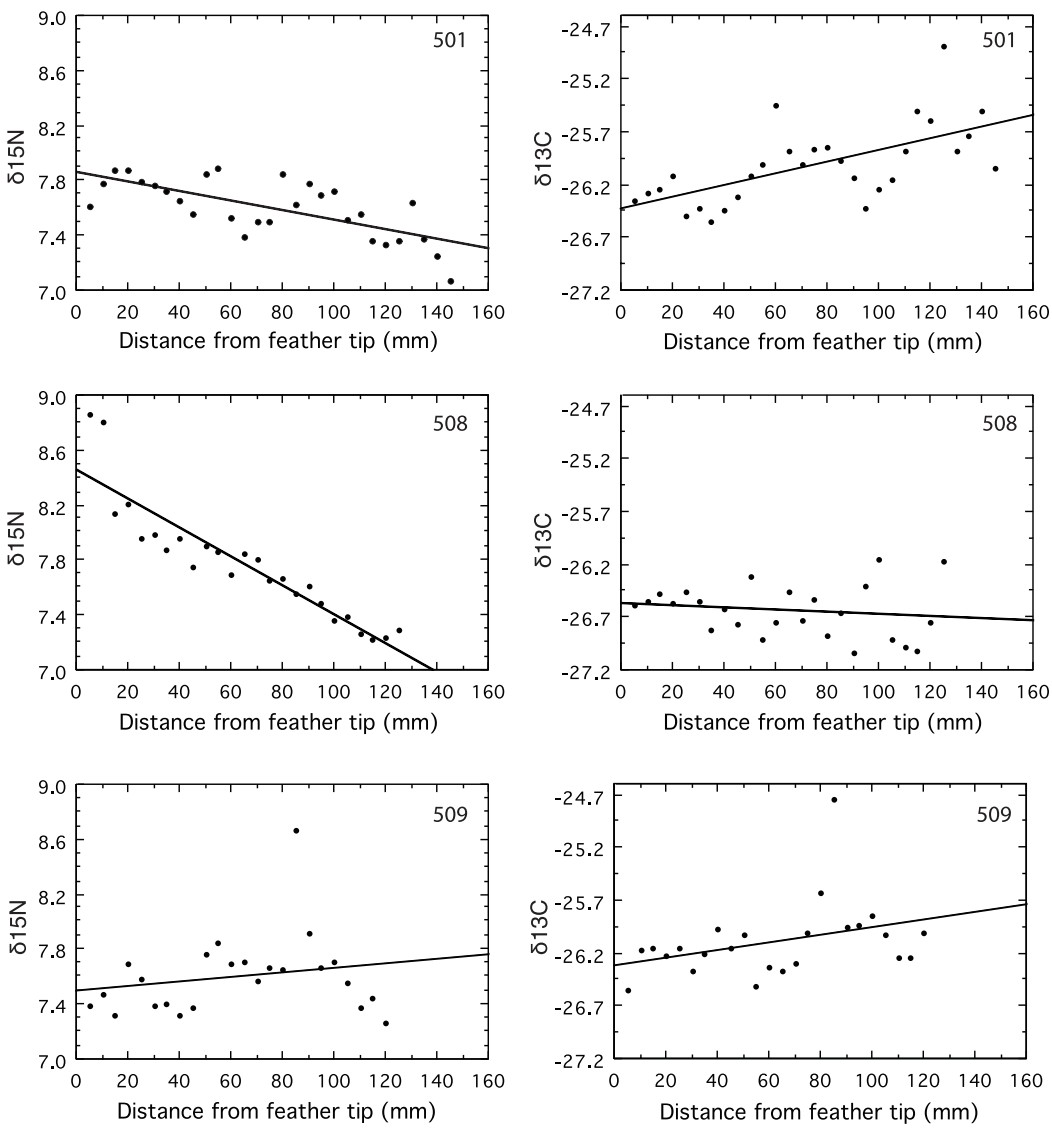

**Figure 1  Values for $\delta^{15}$N and $\delta^{13}$C measured in serial samples along the length of the primary.** Feather vein was sampled at 5 mm intervals near the rachis of the growing primary for its full length, starting at the tip of the feather.

for goose 501 strongly support its being a molt-migrant to the island of Saltholm because freshwater and marine signatures for $\delta^{15}$N are very different; yet, its $\delta^{13}$C increased through time (Fig. 1), contrary to expectation from the results of *Fox, Hobson & Kahlert (2009)*.

## Evidence for 24 h cycling in $\delta^{15}$N

Flightless Greylag Geese forage at night on Saltholm (*Kahlert, Fox & Ettrup, 1996*). Thus, we predicted and found a 24-hour periodicity in the $\delta^{15}$N signature from chronological found samples of the two geese (501, 508) coming into equilibrium with the Saltholm marine environment (Fig. 2). In contrast, N was constant along the length of the primary

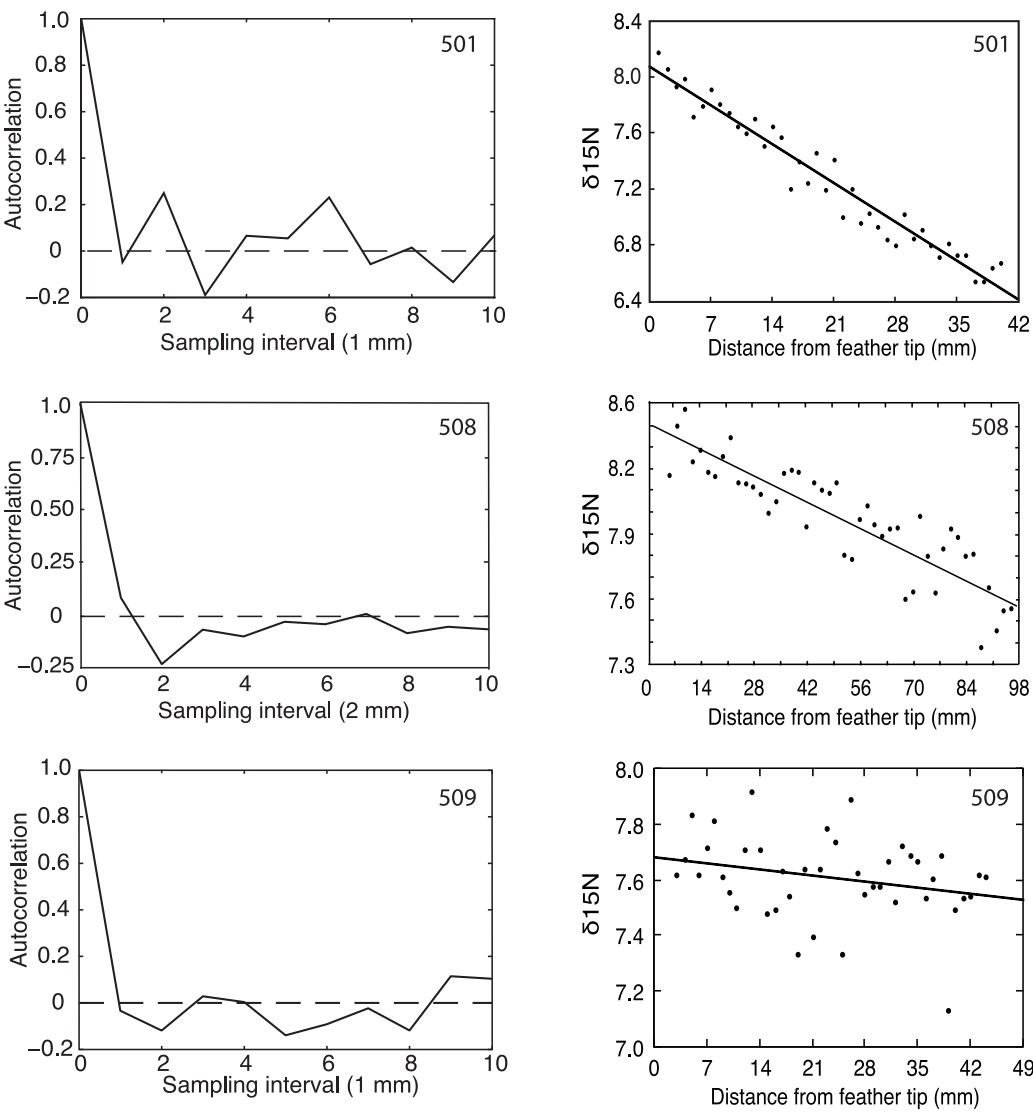

**Figure 2 Autocorrelation and regression results for $\delta^{15}$N measured at 1 or 2 mm intervals from the tips of growing primaries.** Greylag Geese 501 and 508 showed decreasing $\delta^{15}$N values, indicating they were equilibrating with the Saltholm isoscape while their primaries were growing, and both showed significant autocorrelations ($p = 0.01$ and $0.03$, respectively). Goose 509 showed no change in its $\delta^{15}$N values and no autocorrelation ($p > 0.25$).

in 509, as expected for a resident goose in equilibrium with its diet. Because of sampling problems and a malfunction of the mass spectrophotometer, these fine-resolution runs covered fewer days than the regression analyses, which should tend to make periodicity in the data harder to demonstrate.

The $\delta^{15}$N signal is not purely periodic in any of the sampled feathers because of sampling noise. Feather 508 shows a minimum (most negative) autocorrelation value at the second autocorrelation lag. Given a sampling interval of 2 mm, this corresponds to 4 mm of feather length. Additionally, 508 showed positive autocorrelation values in the

region of twice the minimum, strongly indicating periodicity in the data. The bootstrap statistics indicated that the probability of having the combination of a minimum at the lag of 2 and a max near the lag of 4 is $p = 0.032$. Feather 501 had a different sampling interval and, accordingly, showed a more expanded autocorrelation function with a minimum at lag of 3 and a maximum autocorrelation at twice that value. The bootstrap probability of having that combined maximum and minimum was $p = 0.01$. Thus, feathers 501 and 508 both had recognizable periodicity in their $\delta^{15}N$ values. In contrast and as predicted, feather 509 that had constant $\delta^{15}N$ through time offered no evidence of a periodic signal ($p > 0.25$; Fig. 2).

## DISCUSSION

### The value of sampling feathers serially

As far as we are aware, the data for $\delta^{15}N$ in Fig. 1 constitute the first direct test of a gradual change in the isotopic composition of feathers being grown while a molt-migrant is coming into equilibrium with a new isoscape. *Fox, Hobson & Kahlert (2009)* inferred this process by sampling food plants used by Greylag Geese on their breeding grounds in southern Sweden, and on their saltwater molting grounds on the island of Saltholm in Denmark. This inference was based on the assumption that fractionation values for the conversion of $\delta^{15}N$ values in food plants to $\delta^{15}N$ values in goose feathers were accurately represented by the results of an experimental study of Japanese Quail (*Coturnix japonica*) raised on a plant-based diet (*Hobson & Clark, 1992a*; *Hobson & Clark, 1992b*). How well those values represent similar processes in Greylag Geese is an unknown, as are the confidence intervals associated with these transformations. Furthermore, *Fox, Hobson & Kahlert (2009)* used only two food plants from each locality to infer the expected changes in $\delta^{15}N$ and $\delta^{13}C$ values for feathers, yet Greylag Geese probably use a larger diversity of plants at each of these localities, as is known for molting Saltholm geese (*Fox, Kahlert & Ettrup, 1998*). The direct measure of change for $\delta^{15}N$ for two of the three Greylag primaries in our sample offers a powerful confirmation of the result obtained by *Fox, Hobson & Kahlert (2009)*. The mean value of 8.4‰ for $\delta^{15}N$ for feathers from 12 molting geese (*Fox, Kahlert & Ettrup, 1998*) is reasonably close to the mean of 7.6‰ for the two geese that were equilibrating with the Saltholm environment. Mean $\delta^{15}N$ for the goose that showed no evidence of equilibrating was also 7.6‰ (509), considerably higher than the latest (most proximal) $\delta^{15}N$ values for the two geese that were equilibrating (Fig. 1). The relatively high mean $\delta^{15}N$ for goose 509, together with the lack of change in $\delta^{15}N$ along its feather, suggests that this goose had not arrived early and delayed the start of its molt until reaching equilibrium with the Saltholm isoscape. Possibly it was a Saltholm resident with a different diet.

*Fox, Hobson & Kahlert (2009)* suggested that $\delta^{13}C$ also changed in a way that suggested the use of endogenous carbon in the generation of primary feathers during the molt. However, the absolute difference in the expected values for $\delta^{13}C$ (again, generated by sampling two food plants from the Swedish breeding grounds and two food plants from the Saltholm molting grounds) was less than 2‰. While mass spectrometers can readily measure differences as small as 2‰, predicting differences this small by applying fraction

values to the $\delta^{13}$C values measured in samples of two food plants consumed by geese at their breeding and molting sites seems hazardous. With their sample of 12 geese, *Fox, Hobson & Kahlert (2009)* did find the feather values to be intermediate between the food values for Sweden and Saltholm, using the conversion figures for Japanese Quail (*Hobson & Clark, 1992b*). Our mean $\delta^{13}$C value of $-26.2$‰ for the three feathers we analyzed is close to their mean of $-26.5$‰ based on 12 geese (*Fox, Hobson & Kahlert, 2009*). However, the results of *Fox, Hobson & Kahlert (2009)* suggest that $\delta^{13}$C values should decline during primary growth but we found no evidence for such a decline: two geese showed no change, while the third showed a significant increase in $\delta^{13}$C along the length of its primary (Fig. 1). Furthermore, goose 501 with increasing $\delta^{13}$C values showed a strong decline in $\delta^{15}$N values along the length of its growing primary, indicating it was not yet in equilibrium with the Saltholm $\delta^{15}$N isoscape. The positive slope for $\delta^{13}$C in this goose further suggests that the expected difference in feather $\delta^{13}$C, estimated from food plants sampled in Sweden and Saltholm (*Fox, Hobson & Kahlert, 2009*), was not reliable.

### Stored reserves and 24-hour cycling

The two geese with declining $\delta^{15}$N along their primaries also showed 24-hour cycling in their $\delta^{15}$N values, as predicted. Furthermore, the goose with no change in $\delta^{15}$N along its primary showed no evidence of 24-hour cycling, which was predicted because it was not coming into equilibrium with the $\delta^{15}$N environment of Saltholm. These results support the use of endogenous reserves for feather growth during parts of the 24-hour cycle when geese do not forage and feathers continue to grow (*Murphy & King, 1990*). The periodicity of this cycling corresponds to primary growth rates of roughly 8 and 6 mm d$^{-1}$ for feathers 509 and 501, respectively. These inferred rates of primary growth accord well with a growth rate of about 7 mm d$^{-1}$ for a bird the size of a Greylag Goose (*Rohwer et al., 2009*).

Although little of biological importance can be concluded from three geese, it is important to emphasize that we could think of no alternative hypothesis that could account for (1) the concordance we found between equilibration and 24 h cycling in the use of exogenous and endogenous protein sources for feather generation, and (2) the period of this cycling matching the expected primary growth rate for Greylag Geese. Finer sampling, that could be achieved with laser ablation, presumably would eliminate the noise in our autocorrelation results resulting from 1 or 2 mm sampling intervals, leaving only noise associated with day-to-day differences in food intake and feeding times (*Moran et al., 2011*).

### General

*Bridge et al. (2011)* assessed the possible use of endogenous protein reserves for molting by studying changes in $\delta$D and $\delta^{13}$C in the primaries of Painted Buntings (*Passerina ciris*). Like many other migrant song birds that breed in the central and southern regions of western North America, Painted bunting from the Midwestern breeding population migrate to northwest Mexico for their annual post-breeding molt (*Thompson, 1991*; *Rohwer, Butler & Froehlich, 2005*; *Rohwer, 2013*). Here they exploit a food flush generated

by the late summer monsoon, which delivers most of the annual precipitation to this region in July–September (*Adams & Comrie, 1997*; *Comrie & Glenn, 1998*). Primary replacement in Painted Buntings in Sinaloa is so rapid that it requires an average of only 30 and 34 days in adult females and adult males, respectively (*Rohwer, 2013*).

*Bridge et al. (2011)* showed that both $\delta D$ and $\delta^{13}C$ values changed from primary 1 to 9 in some Painted Buntings sampled in Sinaloa. They suggest that birds with differences between primaries 1 and 9 should be individuals that had initiated molt shortly after arriving in Sinaloa, before their endogenous protein reserves reached equilibrium with the Sinaloa isoscape. Individuals without strong differences between these primaries either may have delayed molt until their endogenous protein reserves were in equilibrium with the Sinaloa isoscape, or the food they consumed before migrating may have matched what they were consuming on their Sinaloa molting grounds. Direct evidence of continuous change in $\delta D$ and $\delta^{13}C$ is needed to test the suggestion by *Bridge et al. (2011)* that those buntings with strong differences in $\delta D$ and $\delta^{13}C$ signatures between primaries 1 and 9 were coming into equilibrium with a new isoscape while molting. This could now be accomplished by sampling across different primaries on the feather-time axis developed by *Rohwer & Broms (2012)*, which spans the replacement of all primaries.

In general, measuring isotopic changes in chronological samples taken at equal intervals from flight feathers offers a powerful tool for studying molt-migration. It provides strong data for individual birds while avoiding the assumptions involved with food sampling and using fraction estimates to compute expected tissue values for isotopes. Serial samples representing equal time intervals through primary growth can be generated in two ways. For large birds, finely spaced samples along the length of a primary are chronologically so accurate that they can be used, not only to assess isotopic change during feather growth, but also to evaluate the expected 24-hour periodicity in isotope measurements driven by foraging schedules. For small birds with short primaries, serial samples from single primaries would generate only a limited temporal series and their primaries grow so slowly (*Rohwer et al., 2009*) that sampling with laser ablation would be required to achieve a sample density sufficient to detect 24-hour cycling (*Moran et al., 2011*). Nonetheless, samples representing approximately equal time intervals across the full primary molt can be taken from different primaries (*Rohwer & Broms, 2012*). Sampling across the full chronology of primary replacement extends the sampling period enough that changes in isotopes during primary growth should reliably identify individual molt-migrants, even in small birds.

## ACKNOWLEDGEMENTS

Thanks to the Danish Forest and Nature Agency for permission to catch and sample geese, the landowners of Saltholm for permission to work on the island, Niels Adamsen for his practical help and Ebbe Bøgebjerg and Jens Peter Hounisen for catching molting geese. Thanks also to Johnny Kahlert for assistance, support and inspiration, to Sarah Engel for serially sampling the feathers, and to Jared Grummer for help with the figures. Comments from James Roper and André Guaraldo helped improve the manuscript. Special thanks to PeerJ for its new model of open access publishing.

### Funding

The Øresund Consortium provided financial support of the original fieldwork. The funders had no role in study design, data collection and analysis, decision to publish, or preparation of the manuscript.

### Grant Disclosures

The following grant information was disclosed by the authors:
Øresund Consortium.

### Competing Interests

The authors declare there are no competing interests.

### Author Contributions

- Sievert Rohwer conceived and designed the experiments, analyzed the data, contributed reagents/materials/analysis tools, wrote the paper, prepared figures and/or tables, reviewed drafts of the paper.
- Anthony D. Fox conceived and designed the experiments, analyzed the data, contributed reagents/materials/analysis tools, reviewed drafts of the paper, contributed the feathers.
- Thomas Daniel conceived and designed the experiments, analyzed the data, contributed reagents/materials/analysis tools, prepared figures and/or tables, reviewed drafts of the paper.
- Jeffrey F. Kelly conceived and designed the experiments, performed the experiments, analyzed the data, contributed reagents/materials/analysis tools, reviewed drafts of the paper.

### Animal Ethics

The following information was supplied relating to ethical approvals (i.e., approving body and any reference numbers):

Anthony Fox obtained special written permission from the Danish Forest and Nature Agency to cannon-net Greylag Geese on Saltholm and sample their feathers.

### Field Study Permissions

The following information was supplied relating to field study approvals (i.e., approving body and any reference numbers):

The geese were caught under the Copenhagen University Natural History Museum Ringing Permit A600 "DMU-Kalø Ringmærkning."

### Supplemental Information

Supplemental information for this article can be found online at http://dx.doi.org/10.7717/peerj.743#supplemental-information.

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
