# Peer review of "Chronologically sampled flight feathers permits recognition of individual molt-migrants due to varying protein sources"

_PeerJ, doi:10.7717/peerj.743_

## Round 0.1 · original submission · Major Revisions

· Academic Editor

Major Revisions

Your paper clearly has some interesting implications, as both reviewers and I note (please read my comments in the PDF, sent under separate cover).

All three of us felt that your manuscript could be much shorter and less verbose. I am the only native speaker of the three of us, so my PDF has better examples of how to write more succinctly.

Specifically, reviewer Guaraldo had some relevant observations about reporting the numbers and justifying their variability and the potential for other explanations for their isotope patterns. I would especially like to see comments to that effect - that is, are there alternative explanations, especially considering the sample size. Reviewer 2 agreed that the study had potential, but I felt that he was a bit confused by the writing style and the several places in which you were somewhat prolix about the details. I suggest you look closely at my comments, that are mostly editorial, and recognize the pattern that I was pointing out (use of the word "showed," for example). I think that considering the limitations of your sample size, you could certainly reduce the text to be less speculative and more direct about the results.

I hope you find all of our comments and suggestions useful.

·

Basic reporting

The manuscript presents a new approach to evaluate chronological variations in bird molting behavior through multiple sampling and stable isotope analyses of primary feathers. Despite the very low sample size (three individuals), the manuscript holds a status of an important piece of work by providing the base experiments and data from a novel approach to guide future research on molting ecology and behavior, an important but still little studied theme. Some aspects still deserve further attention from authors to improve this great piece of work, and therefore I provide to the authors a list of specific comments that should help them in this task.

Experimental design

No Comments.

Validity of the findings

No Comments.

Additional comments

Line 153: Such δ15N variations seems very low (~1‰) to be biologically/ecologically meaningful. I question myself to how strong this data is to support the statement that such shifts are exclusively due to environmental change (i.e., distinct molting sites) and not to other sources of variation. For instance, I can imagine that shifts could also be due to individuals' stress, to the unpredictable effect of environmental temperature variations on birds physiology/metabolism, and even due to natural variation of δ15N in a given food item pool and individual variation of foraging strategy or even diet. Maybe this could be better explored together with the statement at line 234.
L. 154-156: Given its importance in this study, I believe that data from Fox et al. (2009) should be clearly presented at Introduction.
L. 163: Mean δ15N of individual 509 is only about 0.4‰ different from 501 and 508 means (respectively, ~7.2‰ and ~7.1‰). Is this difference great enough to assume individuals indeed used distinct feeding grounds? This should be clear to readers (maybe at Introduction section?). Moreover, I noticed some inconsistency in this regard (Discussion, lines 226-228). Authors consider that individuals 501 and 508 are good representatives of a previously sampled Greylag population despite their mean δ15N values differ on about 0.8‰. Therefore, it is not clear to the reader what a reasonable (ecological) difference in δ15N is. This inconsistency follows on lines 229-231 referring to the individual 509.
L. 163-165: δ15N variation between all three individuals is very low in an ecological perspective. Is it possible to infer birds were feeding at distinct environments based only on this slight difference? Maybe if authors present detailed info on local δ15N ratios, readers would better follow and accept their rationale.
L. 168: Although the regression is statistically significant, I think it is more important that authors interpret the ecological meaning of a <2‰ variation in δ13C. At this point, it should be clear what this variation represents in terms of where birds were molting, but this is explained to readers only in Discussion (L. 235-242). Moreover, mean values for 501 and 509 seems very similar (~-25.9‰), an aspect never stressed as meaningful by authors.
L. 172-174: Again, I beg to differ that a statistical significance means distinct ecological scenarios, and this is noticeable when authors justify their findings at Lines 254-257. Moreover, I would move these ideas (starting on line 170) to the Discussion section rather than including them in Results.
L. 182: Check for typos (“indiviuduals”). I may have missed other typos and punctuation errors along the text.
L. 185-188: I think authors could make a brief statement on the extent that the “incomplete” analysis of feathers could affect their results.
L. 236: Authors should consider using “carbon” instead of “C”. In addition, it seems that a line break was mistakenly included in here.
L. 245 and 246: I believe that δ13C values are all negative. Check for missing “minus” signals.
L. 249: I suggest changing “primary growth” by “primary feather growth”.
L. 260: A comma is missing in here: “As predicted, (…)”.
L. 272-273: Even though I have re-read Methods, I could not follow the equipment limitations highlighted here by the authors.
Fig. 1: I missed the equations of each regression line.

Please, check for typos in the author's names. I believe the last author's name is "Jeffrey" instead of "Jeffery".

Reviewer 2 ·

Basic reporting

No additional coments

Experimental design

No additional comments

Validity of the findings

No additional comments

Additional comments

The main idea of the authors is interesting but the manuscript could be more appropriately focused on the main issues (the methods). In my opinion there are too many sentences out of place (i.e. results and conclusions in the other sections of the paper, since the introduction), which makes the paper more speculative unnecessarily. Thus, I believe the paper could be much shorter and direct showing the same conclusions more clearly, avoiding repetitive sentences in every section. My main suggestion is that authors should address the lack of sampled individuals as an a priori statement (more than what they tried to do, only commenting on it in the end of the introduction), and then they should rearrange the text to focus even more on the methodological advance rather than in biological hypothesis based on a sample of only three geese.

---

## Round 0.2 · accepted · Accept

· Academic Editor

Accept

This version shows you attended to most or all of the reviewer's comments. I also agreed with your replies to many of the reviewer's comments in your rebuttal. Finally, while I am accepting your manuscript, there were a few minor places in the text, and the title, that I would like for you to consider. I will communicate this to the editorial staff as well, and annex your PDF with those few comments.